# Validation of a New Duplex Real-Time Polymerase Chain Reaction for *Chlamydia trachomatis* DNA Detection in Ocular Swab Samples

**DOI:** 10.3390/diagnostics14090892

**Published:** 2024-04-25

**Authors:** Joana da Felicidade Ribeiro Favacho, Keren Kariene Leite, Thiago Jacomasso, Aline Burda Farias, Luciano Chaves Franco Filho, Samara Tatielle Monteiro Gomes, Herald Souza dos Reis, Gardene Dourado Mota, Pedro Henrique de Caires Schluga, Walleyd Sami Tassi, Rita de Cássia Pontello Rampazzo, Sheila Kay West, Charlotte Ann Gaydos, Antonio José Ledo Alves da Cunha, Alexandre Dias Tavares Costa

**Affiliations:** 1Evandro Chagas Institute, Secretariat of Health and Environment Surveillance, Ministry of Health (IEC/SVSA/MS), Ananindeua 67030-000, PA, Brazil; lucianofranco6@gmail.com (L.C.F.F.); herald.reis@live.com (H.S.d.R.);; 2Institute of Molecular Biology of Paraná (IBMP), Curitiba 81350-010, PR, Brazilthiago.jacomasso@ibmp.org.br (T.J.);; 3Dana Center for Preventative Ophthalmology, Johns Hopkins University, Baltimore, MD 21287, USA; shwest@jhmi.edu; 4International Sexually Transmitted Disease Research Laboratory, Division of Infectious Diseases, Johns Hopkins University, Baltimore, MD 21218, USA; 5Institute of Studies in Public Health, Federal University of Rio de Janeiro, (IESC/UFRJ), Rio de Janeiro 21941-592, RJ, Brazil; 6Carlos Chagas Institute, Fundação Oswaldo Cruz (FIOCRUZ PR), Curitiba 81350-010, PR, Brazil

**Keywords:** trachoma, *Chlamydia trachomatis*, diagnosis, qPCR, direct immunofluorescence

## Abstract

Trachoma is the world-leading infectious cause of preventable blindness and is caused by the bacteria *Chlamydia trachomatis*. In developing countries, diagnosis is usually based on clinical evaluation. Serological-based tests are cheaper than molecular-based ones, but the latter are more sensitive and specific. The present study developed a new duplex qPCR which concomitantly detects the *C. trachomatis* cryptic plasmid and the human 18S rRNA gene, with an LOD95% for *C. trachomatis* DNA of 13.04 genome equivalents per reaction. The new qPCR was tested using 50 samples from an endemic area and 12 from a non-endemic area that were previously characterized using direct immunofluorescence assay (DFA) and clinical evaluation. Among the 50 endemic samples, 3 were found to be positive by clinical evaluation (6%), 18 were found to be positive by DFA (36%), and 48 were found to be positive by qPCR (96%). Next, the new duplex qPCR was validated using 50 samples previously characterized by qPCR. Validation was carried out on a benchtop instrument (ABI7500) or on a portable point-of-care instrument (Q3-Plus), showing 95% specificity and 100% sensitivity. The ubiquitous presence of *C. trachomatis* DNA in samples from the endemic region confirms that constant monitoring is of paramount importance for the effective measurement of the elimination of trachoma. The newly developed duplex qPCR presented in this study, along with its validation in a portable qPCR system, constitutes important tools toward achieving this goal.

## 1. Introduction

Trachoma is a chronic infectious disease characterized by repeated infections of the ocular conjunctiva by the bacteria *C. trachomatis*. It is one of the leading causes of preventable blindness in developing countries, being associated with poor infrastructure and hygiene conditions (www.trachomacoalition.org accessed on 15 December 2023). In Brazil, trachoma is present in most of the country, but especially in low-socioeconomic-development areas [1].

*C. trachomatis* has 19 serovars, each with distinct characteristics, and trachoma is caused mainly by serovars A, B, Ba, or C [2]. Clinical symptoms include acute inflammation of the conjunctiva and cornea, high mucus production, presence of follicles, and reduced sight [3]. Depending on the intensity, the infection course can either resolve spontaneously or cause tissue fibrosis [4,5], resulting in trichiasis, damage to the cornea, and, ultimately, blindness after repeated infections [4,6]. In Brazil, trachoma diagnosis still relies on direct clinical evaluation [1]. Also, although a gold-standard method of laboratory testing has not yet been defined, some protocols allow for laboratory-based diagnostics to confirm of the condition.

Serological methods are more common than molecular-based methods, mainly due to the easiness of use and relatively lower cost, but each method has its own set of advantages and disadvantages. Direct immunofluorescence assay (DFA) has been used in some Brazilian diagnostic laboratories in recent decades, although qPCR has become more prevalent due to its greater sensitivity. DFA uses either monoclonal antibodies to detect the presence of a *C. trachomatis* specific protein, MOMP, or polyclonal antibodies against lipopolysaccharide (LPS) [7]. However, despite displaying high specificity, DFA’s sensitivity is low [8,9,10,11], possibly because trachoma sampling is affected by several factors, such as recent previous use of eyewash by the patient, associated infections, or even the stage of the infection. Micro-immunofluorescence methods use serovars-specific monoclonal antibodies, which increase the sensitivity for *C. trachomatis* detection, but this is a laborious technique that requires a trained technician to analyze the data, thus limiting its use [12].

Molecular-based protocols are faster, more sensitive, and more specific than serological-methods for detecting *C. trachomatis*, real-time PCR (qPCR) being the most common [11]. Several qPCR protocols have been published, mostly targeting urogenital infection [13,14,15,16,17]. Fully or semi-automated commercial tests aiming to detect the urogenital infection such as the GeneXpert CT/NG (Cepheid, Sunnyvale, CA, USA), the Aptima Combo2 CT/NG assay (Hologic, Marlborough, MA, USA), or the Amplicor CT/NG (Roche Molecular Systems, Pleasanton, CA, USA) have been shown to be very sensitive and specific for the diagnosis of trachoma [15,18,19,20,21,22]. These tests require a clinical laboratory to implement, and trachoma is prevalent in areas with low access to healthcare services and a lack of basic infrastructure, impairing the use of such sensitive instruments [21,23,24,25].

The current diagnostic strategy, which involves determining the prevalence of active trachoma through population-based surveys and acting based on the results, has made significant strides toward trachoma elimination [26]. However, considering the growing concern regarding antibiotic resistance, mass treatment of whole districts with antibiotics (the “A” in the “SAFE” strategy) might not be seen as the best choice, especially after the advent of point of care (POC) testing which allows for in loco decisions of treatment. While rapidly transporting the samples to a fully equipped laboratory in a major city partially solves the issue [15,24,25,27], it does not allow for the immediate start of treatment. The ability to immediately start treatment is regarded as a major advantage of POC tests as patients do not have to return to obtain the results of the test and start treatment [23]. Several true POC assays have been evaluated for detection of ocular *C. trachomatis* infections with important results [28,29,30,31], using several approaches, such as immunochromatographic lateral flow tests [32,33,34], isothermal nucleic acid amplification techniques [35], qPCR [21,36], or electrochemical detection [37], with varying limits of detection (LODs) and specificity/sensitivity values. Such differences might be attributed to the distinct target molecule and intrinsic differences between the techniques. An ideal test, however, should be able to detect a single bacterium of *C. trachomatis* if present in the sample, setting the desired lower limit of detection at 1 genome (which contains about 10 copies of the cryptic plasmid [38]).

The present study aimed to develop and validate a new research duplex qPCR reaction for concomitant detection of *C. trachomatis* cryptic plasmid and the human 18S rRNA gene in ocular swab samples, tailored for the needs of the Brazilian national health system in terms of equipment, quality, and cost, without dismissing sensitivity and specificity. We used 50 pre-characterized ocular swab samples to compare the performance of the new qPCR to the performance of a commercial test (Aptima Combo2 CT/NG assay) and found similar results. Next, the new qPCR was used to re-evaluate 50 samples from an endemic Brazilian area that were previously characterized by DFA. This comparison provides direct experimental evidence to support the use of qPCR to diagnose trachoma in the Brazilian public health system. Lastly, we validated the new duplex reaction in a portable qPCR instrument, the Q3-Plus system [39], in a first step towards the application of the test in field settings.

## 2. Materials and Methods

### 2.1. Samples

Different sample sets were used. Commercial DNA *Chlamydia trachomatis* DNA serovar D (cat# VR-885D) and serovar J (cat# VR-886D) were purchased from ATTC (Manassas, VA, USA) and were used in all experiments except for determination of LOD95%. For LOD95% determination, an analytical standard DNA with a certified number of genome copies was purchased from Vircell Microbiologists (cat# MBC012 lot# 19MBC012101-C01, Granada, Spain). Then, a standard curve was produced in each run as a positive control to validate the results. Partial sequences of the analytical standard DNA can be found at GenBank with accession numbers DQ231369.1, KP120855.1, and JX569833.1. Lyophilized DNA was diluted in 100 µL of molecular-grade water following the manufacturer’s instructions, resulting in a concentration of 18,000 genomes/µL for the product from Vircell and 10 ng/µL for the product from ATCC. Stock solutions were aliquoted and stored at −80 °C until use. Each commercial DNA was 1:10 serially diluted in human trachoma-negative extracted DNA for the investigation of the dynamic linear range detection. Johns Hopkins University’s patients’ samples: Fifty samples were collected using dry swabs from everted upper eyelids of trachoma patients in Tanzania [25,27], which were then characterized for the presence of *C. trachomatis* DNA using the Aptima Combo2 CT/NG assay (Hologic Marlborough, MA, USA), and saved in the biorepository at the Johns Hopkins University’s Chlamydia Research Laboratory (Baltimore, MD, USA). Instituto Evandro Chagas’ patients’ samples: Fifty samples were collected in an endemic area (Marajo Island, Brazil and Ananindeua, Brazil) and twelve were collected in a non-endemic area (Curitiba, Brazil). Each upper eyelid was everted and analyzed by a nurse trained in the WHO simplified classification scheme, who collected samples of individuals irrespective of clinical signs of trachoma. Samples were placed in the specific collection tube, according to the manufacturer’s instructions (digene^®^ Female Swab Specimen Collection Kit, Qiagen, Germamtown, MA, USA). Trachoma-negative human DNA was extracted from swabs collected from everted upper eyelids of non-endemic area volunteers, aliquoted, and stored at −80 °C until use. Volunteers of the non-endemic area were selected among families with high income and education levels, further minimizing the risk for *C. trachomatis* presence [40,41]. On both sites, efforts were made to avoid cross-contamination during sample collection. Efforts included frequent hand washing, change of gloves between each examinee, and ensuring that the swab was placed directly into the collection tube after it touched the conjunctiva, sealing the collection tube without any further contact. A synthetic double-strand DNA molecule (IDT, Coralville, IA, USA) containing one copy of the sequence targeted by the new reaction was used to calculate the LOD95% in absolute copy numbers using serial dilutions of a stock solution of known concentration for the portable thermocycler Q3-Plus. A synthetic positive control DNA was included in each run to validate the results.

### 2.2. DNA Extraction

Archived de-identified patient samples from Johns Hopkins University or Evandro Chagas Institute were vortexed to homogenize the sample collection gel. Two hundred microliters of the gel were aliquoted into a clean tube for processing with QIAamp DNA Blood Mini kit (Qiagen, Germamtown, MA, USA). Samples were also processed with the DNA extraction kit High Pure PCR Template Preparation kit (Roche Applied Sciences, Penzberg, Bavieira, Germany), and results were indistinguishable (see Section 3 and Appendix A). Extracted DNA was aliquoted and stored in a −80 °C freezer. Integrity and concentration of extracted DNA as well as protein contamination were evaluated spectrophotometrically on a Nanodrop 2000c (Thermo Scientific, Waltham, MA, USA).

### 2.3. qPCR Conditions

Detection of *C. trachomatis* cryptic plasmid DNA [42] was performed using the Multiplex PCR Mastermix (IBMP, Curitiba, Brazil) containing Taq DNA polymerase, 9 mM Mg-acetate, 0.8 mM dNTP (Thermo Scientific, Waltham, MA, USA), 5 µL of extracted DNA (1 µL in case of the portable qPCR instrument), oligonucleotides cryp05F (0.2 µM, 5′GGCGTCGTATCAAAGATATGG3′), cryp06R (0.2 µM, 5′CGATGATTTGAGCGTGTGTA3′), and cryp11P (0.1 µM, 5′FAM-TCTCGGGTTAATGTTGCATGATGCTT-BHQ1 3′) in a reaction volume of 25 µL (5 µL in the case of the portable qPCR instrument). Specific oligonucleotides were designed for concomitant detection of human 18S rRNA gene (NR_145820.1). Hence, the reaction also contained the oligonucleotides 18S2F (0.1 µM, 5′GAAACTGCGAATGGCTCATTAAATCA3′), 18S2R (0.1 µM, 5′AGAGCTAATACATGCCGACGGG3′), and 18S2P (0.05 µM, 5′HEX- TGGTTCCTTTGGTCGCTCGCTCC-BHQ13′). The complete reaction mix, including primers and probes, was produced by IBMP (Curitiba, Brazil) under current Good Manufacturing Practices (GMP). Primers and probes were purchased from IDT (Coralville, IA, USA), and were purified by reverse phase and HPLC, respectively. All six oligonucleotide sequences were tested for unspecific reaction against 55 common parasites and bacteria, but no amplification was observed (a list of the organisms is shown in Appendix A). Alignment of the targeted sequence of the cryptic plasmid from all serovars is shown in Appendix A. The location of the targeted sequence on the plasmid, ORF3, is shown is Appendix A (blue segment on red arrow). Reactions were performed and analyzed on the ABI7500 Standard instrument (software v2.0.6, Thermo Fisher Scientific—Waltham, MA, USA), with the following cycling conditions: 95 °C/10 min, 45 × [95 °C/15 s + 60 °C/1 min], and with ROX as passive fluorescence reference. The baseline was set from 3 to 15 for both targets. For *C. trachomatis* DNA, the threshold was set to 0.1, and quantification cycle (Cq) values between 19 and 42.50 were considered positive detections. For human DNA, the threshold was set to 0.1 and detections were considered positive and free of contaminants if Cq values were between 14 and 37. The reactions on the portable Q3-Plus system used the following cycling conditions: 97 °C/60 s, 45 × [97 °C/20 s + 64 °C/1 min], and with no passive fluorescence reference. Optical parameters for FAM channel (*C. trachomatis* target) were 0.5 s of exposure time, analog gain of 14, and LED power of 7. The baseline was automatically set. Optical parameters for HEX/VIC channel (human target) were 2 s of exposure time, analog gain of 16, and LED power of 10. For *C. trachomatis* DNA, threshold was set to 16, and quantification cycle (Cq) values between 22 and 39 were considered positive detections. For human DNA, the threshold was set to 19 and the detections were considered positive and free of contaminants if Cq values were between 14 and 32. Cq values for the human control target out of specified ranges were interpreted as an inhibition of the reaction or the failure of DNA extraction, and DNA samples were re-extracted. Non-template controls (NTC, molecular-grade water) as well as trachoma-negative human DNA were routinely used as controls. Non-template controls should show no amplification for both targets, whereas trachoma-negative controls should show positive detections for human DNA with Cq values were between 14 and 37.

### 2.4. Direct Immunofluorescence Detection (DFA)

Samples were methanol-fixed in glass slides, transported at 4–8 °C and stored at −20 °C until the analyses were conducted. DFA was performed according to the manufacturer’s instructions of the kit “Chlamydia T” (Biocientífica S.A., Buenos Aires, Argentina). Briefly, fixed samples were exposed to fluorescent-labelled monoclonal antibodies against anti-MOMP from *C. trachomatis*. Slides were analyzed in a fluorescence microscope and were considered valid when containing a minimum of 100 cells and 5 clearly visible Chlamydia antigenic structures. Positive and negative controls were performed in all DFA assays.

### 2.5. Statistical Analysis

All qPCR assays in the ABI7500 system were performed in technical triplicates, except for limit of detection (LOD) determination (8–12 replicates). All qPCR assays in the Q3-Plus system were performed in technical duplicates, except for limit of detection (LOD) determination (4–8 replicates). Results are expressed as mean ± standard deviation of the quantification cycle (Cq) values. Paired Student’s *t*-test (95% confidence level) results were calculated using GraphPad Prism v5.0 software (Boston, MA, USA). The 95% limit of detection (LOD95%) for detection of the Chlamydia genomic target was calculated by Probit regression analysis [43]. Kappa coefficient was calculated between the following: (i) the qPCR results obtained with DNA extracted with the two commercial kits (Qiagen’s Blood Mini kit and Roche’s High Pure PCR Template Preparation kit); (ii) the results obtained with the new duplex qPCR and the results of the pre-characterization of the same samples by the commercial Aptima Combo2 CT/NG assay (taken as the gold standard); (iii) the new duplex qPCR and the DFA assay or the clinical evaluation; (iv) the results obtained with the laboratory instrument ABI7500 and the portable instrument Q3-Plus. The coefficient was used to test agreement between the diagnostic methods, and Kappa results were interpreted according to the following [44,45]: 1.00–0.81—almost-perfect agreement; 0.80–0.61—substantial agreement; 0.60–0.41—moderate agreement; 0.40–0.21—fair agreement; ≤0.20—poor agreement. STARD guidelines were followed in the course of this work. An MIQE checklist is provided in Appendix A.

## 3. Results

### 3.1. qPCR Optimization and Analytical Parameters

A new duplex qPCR was developed targeting a sequence of the cryptic plasmid of *C. trachomatis* and a sequence of the human 18S rRNA gene. A 120 bp target sequence of the cryptic plasmid was chosen because it can be present in up to ten copies per bacterial genome [38,42] and is highly conserved, sharing 100% homology among all serovars (Appendix A), in accordance with previous studies [46].

The performance of the new qPCR was initially developed using commercial genomic DNA extracted from *C. trachomatis* DNA serovars D and/or J, since there are no significant differences in the target sequence among all serovars (Appendix A). The linear dynamic range for detection of the *C. trachomatis* is 0.1 fg/µL to 10^4^ fg/µL when serovar J is diluted in human DNA to mimic a clinical sample (Figure 1). Representative traces of the reactions used to calculate the linear regression (red lines a–f) obtained with concomitant detection of the human 18S rRNA gene (green lines) are shown in the insert in Figure 1. Similar results were obtained using twice the concentration of oligonucleotides (that is, cryp05F at 0.4 µM, cryp06R at 0.4 µM, and cryp11P at 0.2 µM) (Appendix A). The reaction parameters for *C. trachomatis* DNA detection on the benchtop equipment were as follows: efficiency of 92% (slope –3.52), R2 of 98.6%, and Y-intercept of −32.79. Table 1 shows the average ± SD of the quantification cycle (Cq) for each serovar J DNA concentration used in Figure 1, and the same data for the portable instrument Q3-Plus.

However, since DNA from ATCC contains an unknown concentration of human DNA, it is not suitable to accurately determine analytical parameters such as LOD95%. Therefore, a commercial, diagnostic-certified DNA standard for *C. trachomatis* with known concentration (from Vircell Microbiologists), a Probit analysis determined an LOD95% of 13.04 genome equivalents per reaction. Furthermore, when a synthetic double strand DNA containing the same genomic sequence was used as template, LOD95% reached 1.61 copies/µL, which translates to eight copies per reaction (Appendix A) and is close to the limit of the technique of one–three copies/reaction [47]. The repeatability of the new duplex qPCR (intra-assay variability) was found to be <8.8% (highest at 0.1 fg/µL) and the reproducibility (inter-assay variability) was found to be <5.8% (highest at 0.03 fg/µL) when assayed by three independent, experienced operators.

### 3.2. Performance on Clinical Samples

The new duplex qPCR protocol was evaluated with real patient samples collected in Brazil that were scored by both clinical evaluation and a DFA protocol that uses a monoclonal anti-MOMP antibody, according to the recommendations of the Brazilian Ministry of Health [1]. A total of 62 samples were collected, with 50 coming from a trachoma-endemic region and 12 from a non-endemic region. Figure 2A shows the qPCR evaluation of 50 samples from a trachoma-endemic region (Marajo Island, Brazil), while Figure 2B shows the qPCR evaluation of 12 samples from a non-endemic region (Curitiba, Brazil). Green lines represent the detection of the human 18S rRNA gene. Blues lines represent the detection of the *C. trachomatis* cryptic plasmid DNA. Figure 2C shows the dispersion of the Cq for each sample detected in Figure 2A plotted over the linear regression of the detection of serially diluted *C. trachomatis* serovar J DNA. It can be observed that the samples have high Cq, highlighting the importance of the linear detection of 0.1 fg/µL (Table 1) and the non-linear detection of up to 0.01 fg/µL (Figure 2C). It should be noted that any Cq out of the linear detection range cannot be reliably quantified with the current reaction; hence, they are shown here solely to illustrate the importance of detection of low DNA concentrations. It can be observed that the specific bacterial DNA sequence can be found in most of the samples from the endemic area, while no detection was observed in the samples from the non-endemic area. *C. trachomatis* DNA could be identified in 48 of the 50 endemic-area samples, displaying 96% positivity. However, only three of these samples were positive for *C. trachomatis* (Table 2) by clinical evaluation, confirming an already-known strong disparity between the two diagnostic techniques.

The results show that samples 1–50 (endemic region) were also primarily negative for immunodetection of MOMP, whereas 48 of those were classified as “Positive” by qPCR (Table 2). This could be because the qPCR has greater sensitivity, can detect earlier/later stage infections that do not show up via immunodetection, or that the immunodetection method is qualitative and prone to errors in user scoring. Samples 51–62 (non-endemic region) were classified as “Negative” for all three diagnostic techniques (clinical evaluation, DFA, or qPCR). There was no correlation between the clinical evaluation, the number of elementary bodies (EBs) as observed by DFA, or the results of the direct immunodetection protocol with the qPCR results. Indeed, Kappa coefficient analysis showed poor correlation between clinical evaluation and qPCR or between DFA and qPCR (less than 0.2 for both cases).

DNA from Brazilian samples was extracted using two different commercial kits (Qiagen’s QIAmp DNA Blood Mini kit or Roche’s High Pure PCR Template Preparation kit) and a direct comparison of the Cq obtained for selected samples using both DNA extraction kits is shown in Appendix A. Appendix A shows the distribution of the Cq differences shown in Appendix A, with a mean difference of −0.8 Cq if favor of Roche’s product, which is not meaningful in this particular assay. The calculated Cohen’s kappa coefficient shows a 98% agreement (k of 0.957) between the results obtained with the extraction kits, which is considered a substantial agreement. Therefore, both kits were considered equivalent to detect *C. trachomatis* DNA using qPCR.

The new duplex qPCR was then validated with samples previously characterized by a distinct commercial PCR-based test, the Aptima Combo2. DNA from fifty samples analyzed by the Aptima Combo2 assay was also evaluated by the duplex qPCR on a benchtop laboratory equipment, the ABI7500 (Table 3). Samples 1–30 were previously characterized as “Positive” and samples 31–50 were characterized as “Negative” for *C. trachomatis* DNA. Both assays yielded the same result for all samples except for one. “Negative” sample (#34) was classified as “Inconclusive” by the qPCR because the reaction failed to detect the human DNA marker, thus invalidating the results. This sample would have to be re-processed in a clinical setting because failure to detect the human DNA might be attributed to degraded DNA template or other problems with sample collection. These data yield a true-positive rate (sensitivity) of 100% (CI95% of 88.43% to 100%) and a true-negative rate (specificity) of 100% (CI95% of 82.35 to 100%), with no false-negative or false-positive detections, and one inconclusive with our duplex qPCR test. The positive predictive value (PPV) and negative predictive value (NPV) were calculated as 100% and 99.8% (CI95% 98.9 to 99.9%), respectively (Table 4). Cohen’s kappa coefficient was calculated to be 0.957, which means that both assays (the Aptima Combo 2 and the new duplex qPCR performed on the ABI7500 instrument) are in almost-perfect agreement (98%).

### 3.3. Application on a Portable qPCR Instrument

Lastly, the new duplex qPCR was optimized and validated for use in a prototype POC qPCR instrument called Q3-Plus. This is a system that controls the temperature on a silicon-based chip, enabling the amplification of any targeted nucleic acid with concomitant detection of the amplification product via fluorescent signals [39]. The reaction parameters for *C. trachomatis* DNA detection on the portable Q3-Plus were as follows: efficiency of 83.6% (slope –3.79), R^2^ of 98%, and Y-intercept of 36.5. The average ± SD of the quantification cycle (Cq) for each concentration of the synthetic positive control used to evaluate the qPCR performance is shown in Table 1. Appendix A (panel A), shows representative traces of the detection of both *C. trachomatis* and human genomic targets using synthetic DNA molecules. Appendix A (panel B), shows the determination of an LOD95% of 79 copies/µL, which translates to 395 copies per reaction, or roughly 40 *C. trachomatis* genomes. Data used for LOD calculation are shown in Appendix A. Next, the previously Aptima Combo2-characterized samples were evaluated in the portable Q3-Plus system, except for sample #34, which gave inconsistent results. All positive samples were characterized as positive for the portable system, while two of the negative samples were characterized as positive (i.e., false positives) (Table 3). It should be pointed out that an additional five “Negative” samples gave positive signals for *C. trachomatis* DNA detection, but since four of the signals were out of the accepted positive range, they were considered “Negative” (samples #32, #33, #46, and #48). Sample #31 was considered a true false positive. These data yield a true-positive rate (sensitivity) of 100% (CI95% of 88.43% to 100%) and a true-negative rate (specificity) of 95% (CI95% of 75.13 to 99.87%), with no false-negative detections but one-positive detection. The positive predictive value (PPV) and negative predictive value (NPV) were calculated as 99.45% (CI95% 96.38 to 99.92%) and 100%, respectively (Table 4). The calculated kappa coefficient between the results obtained by the commercial test Aptima Combo 2 and the new duplex qPCR in the portable instrument Q3-Plus was found to be 0.917, which is an almost-perfect agreement (96%).

## 4. Discussion

The work presented here shows the development and validation of a new duplex qPCR test for detection of *C. trachomatis* DNA in ocular samples using a benchtop and a portable instrument. The reaction concomitantly detects a human housekeeping gene, the 18S rRNA, bringing reliability to results indicating negative detection of bacterial DNA. Detection of a host gene also controls for the reagents’ quality and adequate instrument operation, which, together with the production of reagents under current good manufacturing practices (cGMPs), ensures the high performance of the newly developed test [43].

The development of the new reaction was performed using commercial DNA, detecting concentrations of *C. trachomatis* DNA corresponding to levels that could be found in ocular samples. More importantly, the sequence targeted by the present reaction is conserved in the cryptic plasmid of all *C. trachomatis* serovars (Appendix A), and is in the ORF3 internal region (Appendix A) to avoid 377-bp ORF1 deletion. The design of new tests for different regions has been shown to be essential since mutations and deletions both in chromosomal regions and in the plasmid have affected the detection efficiency of some commercial kits [48,49,50]. Other PCR protocols, either nested, real-time, or digital, manual, or semi-automated, have also been published in recent years [13,14,15,16,18,19,51,52]. Although most were not developed for ocular samples, they have been validated for such uses, with sensitivity and specificity like those of the original sample matrix (i.e., urovaginal fluids). This is a possible explanation for their limit of detection being around 10 genomes of *C. trachomatis*, which is not the most suitable LOD for the detection of these bacteria in ocular samples [38]. However, much as the reaction presented herein, some published tests do reach the necessary sensitivity of 1 genome of *C. trachomatis* to be able to detect its low abundance in ocular swab samples [14,15,51].

The new duplex qPCR was validated using ocular samples previously characterized for the presence of *C. trachomatis* DNA by the Aptima Combo2 CT/NG assay, which aims for the same genomic target [25,53]. The results reached almost-perfect agreement as per the calculated kappa coefficient, with 98% of correctly identified samples. On the other hand, when the qPCR results were compared to the clinical evaluations or DFA, the agreement was considered poor very likely because of the poor sensitive and qualitative nature of the clinical evaluations and the DFA assay. Clinical examination and laboratory tests are frequently discordant, possibly due to infection kinetics [27] and the age-dependent manifestations of infection [42]. The gold-standard method for the diagnosis of *C. trachomatis* is still, in many countries, clinical evaluation by inspection of the patient’s everted eyelid for lesions [54,55], while some use DFA [9,10,11]. Very few diagnostic facilities perform molecular tests for trachoma determination due to the low socioeconomic level of regions that have trachoma. Thus, given the distinct nature of each test, it is important to identify and differentiate each of the tests by their requirements and biological targets. Clinical evaluation measures the number of lesions in the everted eyelid of both eyes, which requires a skilled health practitioner but few expensive tools. DFA uses antibodies to recognize the presence of *C. trachomatis* proteins in swab samples of the everted eyelids. DFA requires a fluorescence microscope, and it is prone to inconclusive results due to the high levels of background fluorescence; thus, a skilled professional is required here, also, to ensure accurate readings. Moreover, the DFA test threshold for positivity of >5 EBs per analysis could be questioned since the presence of EB is indicative of cellular adaptation for infection [56]. Molecular tests such as qPCR require a thermocycler to detect and amplify sequences of the bacterial genome in a very specific pattern; thus, they are more reliable than DFA. Therefore, despite recent advances, DFA still is less sensitive than qPCR [8,9,10,11,53,57,58,59,60]. Indeed, since the clinical signs of trachoma infection are not so unique [61], clinical evaluations correlate poorly with PCR detection levels in low-prevalence areas or after mass antibiotic treatments [62].

Our results corroborate the differences between all three techniques. When 62 uncharacterized samples were explored for the presence of *C. trachomatis* antigens (DFA) or for *C. trachomatis* DNA (qPCR), almost opposite results emerged (Table 4). Clinical evaluation or detection of EB by DFA showed very few positives amongst the endemic samples, while 96% were found to be positive by qPCR (48 out of 50 samples). This means that, although the patients did not show clinical signs of active infections or scars from repetitive infections, it may be that the post-infection with the bacterial DNA was still present, possibly in tissue that was not yet removed by the body; alternatively, they might be in an early, asymptomatic stage of infection. Although the high positivity found by qPCR could arise from systematic errors resulting in cross-contamination during sample collection and/or processing, the negativity of samples #2 and #41, among of the first and last samples to be collected and processed, testifies to the validity of the sample collection process and the positive qPCR results. We believe that the observed high positivity rate is due to the target population, i.e., schoolchildren. Children are more likely to not wash their hands as frequently as is necessary and to share their belongings with multiple friends, facilitating the spread of *C. trachomatis* in the community. In agreement with our results, Bailey and colleagues found that clinically negative subjects who were PCR-positive were more likely than PCR-negative subjects to have acquired signs of disease at 1 and 6 months of follow-up, while clinical signs were twice as likely to have resolved after 1 month in PCR-negative subjects with disease than in those who were PCR-positive [13]. Indeed, latent class analyses of clinical examination versus qPCR detection suggest that qPCR positivity is a better predictor for determining chlamydial infection than clinical inspections [63].

Importantly, considering that trachoma afflicts populations with low access to healthcare and, in several circumstances, little-to-no access to transportation to diagnostic centers, we also evaluated the new duplex qPCR in a portable qPCR system. The evaluated instrument, known as the Q3-Plus system, is a robust and simple-to-use portable platform [39] that has been shown to accurately detect DNA from protozoan parasite’s *Trypanosoma cruzi* or *Plasmodium* spp. in human blood samples [64] and to correctly genotype acute coronary syndrome patients to provide a personalized approach to selecting antiplatelet therapy and avoid bleeding events [65,66]. In this portable platform, the new qPCR was able to accurately detect *C. trachomatis* DNA in 96% of the pre-characterized samples, even if exhibiting a non-ideal LOD95% of 79 copies/µL. Although this is an important step towards a diagnostic solution that can reach populations that cannot access healthcare centers, the results are similar to those shown by isothermal molecular POC tests, LAMP being the most common. LAMP-based amplification of *C. trachomatis* DNA was evaluated in several configurations, such as coupled to a lateral flow assay [67,68] or in microfluidic chips [69,70]. Although not yet tested with clinical samples or field settings, a recently published study performing real-time digital LAMP assay followed by high-resolution melting analysis showed an LOD of 1–2 copies/μL, which is the same range as other molecular POC assays [71].

Lastly, the limitations of the qPCR technique must not be understated. Molecular assays for trachoma diagnosis might present a false-negative rate as high as 20% [63], possibly because of the infection timing or poor DNA extraction efficiencies [72,73,74,75]. However, our test could possibly eliminate some of these false-negative results with the addition of the internal human control of the 18S rRNA gene. This would identify samples with poor DNA quality and quantity due to DNA degradation or low extraction efficiencies. Second, the assay presented herein relies on the presence of the cryptic plasmid inside the bacteria, which could be missing [76,77]. However, plasmid-free infections are highly attenuated in mouse models [78,79] and plasmid-free *C. trachomatis* serovar A organisms no longer cause pathology in nonhuman primate ocular tissues [80]. Therefore, it is suggested that there is a selective pressure related to infectivity that maintains the cryptic plasmid in naturally occurring infections [80,81], supporting its use as target for diagnostic tests. Accordingly, a detailed analysis of clinical samples has shown that plasmid-free variants of *C. trachomatis* may have an incidence lower than 0.5% [78]. Some authors have also suggested that other species of Chlamydia can also cause trachomatous inflammation [82], although these infections were also found to be attenuated if the plasmid is lacking [80]. Other limitations that were not evaluated in our study include differences between conjunctival and epithelial specimens, human conjunctival cell yield, DNA extraction efficiency, and thorough removal of molecular inhibitors that may also affect test performance [72,73,74,75].

## 5. Conclusions

This work shows the development and validation of a new duplex qPCR that concomitantly detects *C. trachomatis* cryptic plasmid DNA and a human endogenous control target using a benchtop and a portable instrument. The newly developed reaction was validated against qPCR-pre-characterized samples, showing high sensitivity and specificity. It was also compared with traditional techniques for trachoma diagnosis, such as the clinical evaluation of the eyelid or direct immunofluorescence assay, showing greater overall performance. The presented qPCR may be an important research tool for monitoring patients and diagnosing trachoma in ocular samples in endemic regions as part of the effective measures for the elimination of the disease. The validation of the new qPCR in a portable instrument is a step towards this goal.

## Figures and Tables

**Figure 1 diagnostics-14-00892-f001:**
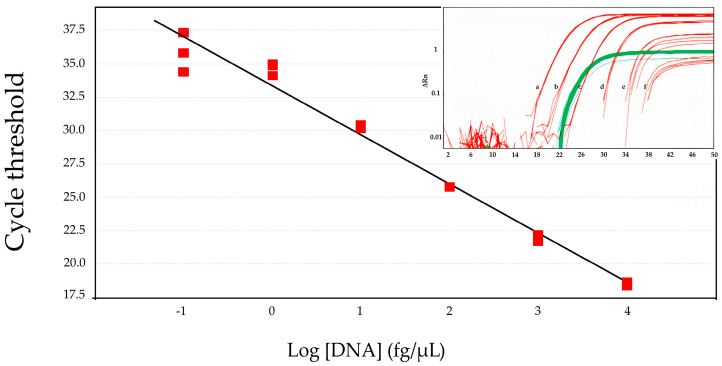
Linear dynamic range for detection of the *C. trachomatis* genomic target in serovar J DNA. *C. trachomatis* DNA was serially diluted in human DNA and evaluated using the new duplex qPCR. The reaction shows efficiency of 92% and R2 of 98.6%. Insert in Figure 1 shows representative traces of the reactions used to calculate the linear regression (red lines a–f, ranging from 1 fg/µL to 10 pg/µL), obtained with concomitant detection of the human 18S rRNA gene (green lines). Traces and linear regression are representative of more than 10 independent experiments with 3–12 replicates per run for each concentration. Samples replicate (red square).

**Figure 2 diagnostics-14-00892-f002:**
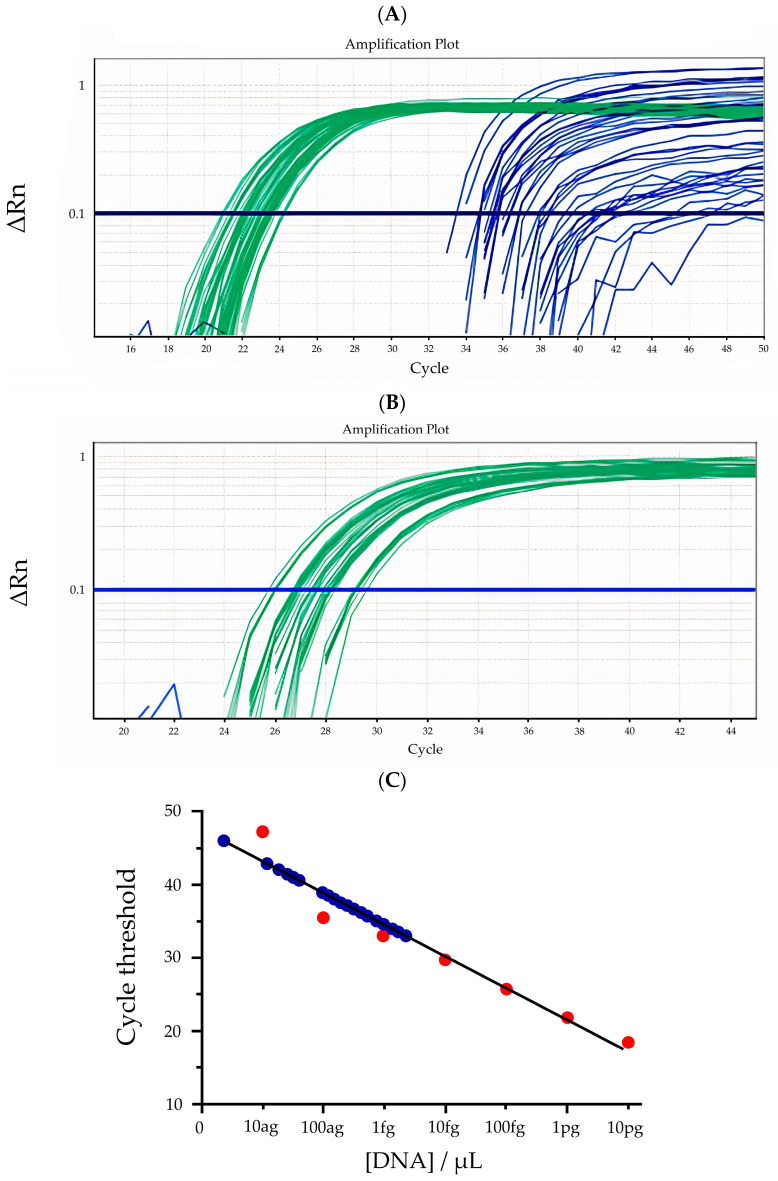
Detection of *C. trachomatis* DNA in human samples. Samples from an endemic and a non-endemic region in Brazil were collected and their DNA extracted as described in Section 2. Panel (**A**) shows the detection of *C. trachomatis* DNA (blue lines) parallel to the detection of human DNA (green lines) in samples from the endemic area. Panel (**B**) shows the detection of human DNA (green lines) but no *C. trachomatis* DNA in samples from the non-endemic area. Panel (**C**) shows the quantification of *C. trachomatis* DNA present in each of the positive samples (blue circles) based on the linear regression obtained with serial dilutions of *C. trachomatis* serovar J DNA (red circles).

**Table 1 diagnostics-14-00892-t001:** Quantification cycle (Cq) for detection of different concentrations of *C. trachomatis* DNA (serovar J) on a benchtop instrument (ABI7500) or of different concentrations of a synthetic positive control on a portable thermocycler (Q3-Plus). Data are shown as mean Cq ± SD and were obtained with 3–5 independent experiments for each instrument. Replicates for each concentration are described in Section 2.

Concentration of *C. trachomatis* DNA (per µL)	Cq (mean ± SD) ABI7500	Number of Copies of the Synthetic Positive Control (per µL)	Cq (mean ± SD) Q3-Plus
10 pg	18.54 ± 0.19	100,000	20.74 ± 0.93
1 pg	22.00 ± 0.29	10,000	23.99 ± 1.31
100 fg	25.99 ± 0.22	1000	26.74 ± 0.95
10 fg	30.14 ± 0.04	100	31.82 ± 1.17
1 fg	34.32 ± 0.65	10	33.92 ± 0.79
0.1 fg	36.76 ± 0.91	1	36.65 ± 1.63

**Table 2 diagnostics-14-00892-t002:** Comparison of duplex qPCR results to the characterization of samples from endemic and non-endemic areas clinical evaluation (CE) or direct immunofluorescence (DFA). The number of elementary bodies (EBs) as well as the qPCR quantification threshold cycle (Cq) observed (using the ABI7500) for each sample is shown.

Sample	Clinical Evaluation (CE)	EBs	DFA Classification	Cq (mean ± SD)	Duplex qPCR Classification	Agreement CE × qPCR	Agreement DFA × qPCR
1	Negative	0	Negative	36.59 ± 0.59	Positive	No	No
2	Negative	8	Positive	-	Negative	Yes	No
3	Negative	0	Negative	36.18 ± 0.54	Positive	No	No
4	Negative	0	Negative	41.62 ± 0.00	Positive	No	No
5	Negative	0	Negative	38.91 ± 3.36	Positive	No	No
6	Negative	0	Negative	35.93 ± 1.49	Positive	No	No
7	Negative	5	Positive	36.06 ± 1.73	Positive	No	Yes
8	Negative	2	Negative	43.05 ± 1.78	Positive	No	No
9	Negative	3	Negative	39.12 ± 2.67	Positive	No	No
10	Negative	0	Negative	42.20 ± 4.22	Positive	No	No
11	Positive	7	Positive	36.82 ± 0.91	Positive	Yes	Yes
12	Negative	2	Negative	38.31 ± 0.00	Positive	No	No
13	Negative	0	Negative	37.89 ± 1.43	Positive	No	No
14	Negative	1	Negative	33.90 ± 0.49	Positive	No	No
15	Positive	6	Positive	36.86 ± 1.49	Positive	Yes	Yes
16	Negative	0	Negative	37.07 ± 2.49	Positive	No	No
17	Negative	2	Negative	37.83 ± 1.46	Positive	No	No
18	Negative	6	Positive	35.08 ± 0.00	Positive	No	Yes
19	Negative	2	Negative	37.61 ± 2.79	Positive	No	No
20	Negative	0	Negative	37.53 ± 2.81	Positive	No	No
21	Negative	2	Negative	33.58 ± 0.89	Positive	No	No
22	Negative	0	Negative	34.55 ± 0.37	Positive	No	No
23	Negative	4	Negative	35.94 ± 0.21	Positive	No	No
24	Negative	5	Positive	34.83 ± 0.16	Positive	No	Yes
25	Negative	0	Negative	37.07 ± 1.46	Positive	No	No
26	Negative	6	Positive	37.05 ± 1.51	Positive	No	Yes
27	Negative	3	Negative	37.78 ± 0.37	Positive	No	No
28	Negative	6	Positive	33.10 ± 0.45	Positive	No	Yes
29	Negative	2	Negative	36.73 ± 0.59	Positive	No	No
30	Negative	5	Positive	37.40 ± 1.19	Positive	No	Yes
31	Negative	6	Positive	35.24 ± 0.30	Positive	No	Yes
32	Negative	7	Positive	35.26 ± 1.04	Positive	No	Yes
33	Negative	5	Positive	34.98 ± 0.76	Positive	No	Yes
34	Negative	6	Positive	34.06 ± 0.00	Positive	No	Yes
35	Negative	8	Positive	35.23 ± 1.08	Positive	No	Yes
36	Positive	9	Positive	38.28 ± 0.50	Positive	Yes	Yes
37	Negative	2	Negative	34.23 ± 0.48	Positive	No	No
38	Negative	2	Negative	35.84 ± 0.84	Positive	No	No
39	Negative	3	Negative	36.93 ± 0.00	Positive	No	No
40	Negative	2	Negative	36.59 ± 0.75	Positive	No	No
41	Negative	5	Positive	-	Negative	Yes	Yes
42	Negative	4	Negative	35.12 ± 1.11	Positive	No	No
43	Negative	5	Positive	40.70 ± 0.00	Positive	No	Yes
44	Negative	7	Positive	34.10 ± 0.77	Positive	No	Yes
45	Negative	3	Negative	34.50 ± 0.36	Positive	No	No
46	Negative	2	Negative	35.23 ± 1.42	Positive	No	No
47	Negative	0	Negative	38.71 ± 1.58	Positive	No	No
48	Negative	1	Negative	33.72 ± 0.50	Positive	No	No
49	Negative	2	Negative	34.53 ± 0.34	Positive	No	No
50	Negative	1	Negative	36.38 ± 0.00	Positive	No	No
51	Negative	0	Negative	-	Negative	Yes	Yes
52	Negative	0	Negative	-	Negative	Yes	Yes
53	Negative	0	Negative	-	Negative	Yes	Yes
54	Negative	0	Negative	-	Negative	Yes	Yes
55	Negative	0	Negative	-	Negative	Yes	Yes
56	Negative	0	Negative	-	Negative	Yes	Yes
57	Negative	0	Negative	-	Negative	Yes	Yes
58	Negative	0	Negative	-	Negative	Yes	Yes
59	Negative	0	Negative	-	Negative	Yes	Yes
60	Negative	0	Negative	-	Negative	Yes	Yes
61	Negative	0	Negative	-	Negative	Yes	Yes
62	Negative	0	Negative	-	Negative	Yes	Yes

**Table 3 diagnostics-14-00892-t003:** Validation of qPCR shown in Figure 1 with pre-characterized patient samples. Samples were pre-characterized by the Aptima Combo2 CT/NG assay (Hologic, USA). Extracted DNA was then analyzed by the NAT Trachoma kit described in the present paper by two different instruments: the ABI7500 and the Q3-Plus.

Sample	Pre- Characterization Status (by Aptima Combo 2)	ABI7500			Q3-Plus		
		Mean Cq for *C. trachomatis* Target	Detection of Reaction Internal Control	Diagnostic Status	Mean Cq for *C. trachomatis* Target	Detection of Reaction Internal Control	Diagnostic Status
1	Positive	22.60	Yes	Positive	23.05	Yes	Positive
2	Positive	30.36	Yes	Positive	29.75	Yes	Positive
3	Positive	29.65	Yes	Positive	28.80	Yes	Positive
4	Positive	25.75	Yes	Positive	25.75	Yes	Positive
5	Positive	30.75	Yes	Positive	29.40	Yes	Positive
6	Positive	39.28	Yes	Positive	36.20	Yes	Positive
7	Positive	38.99	Yes	Positive	36.65	Yes	Positive
8	Positive	32.56	Yes	Positive	32.45	Yes	Positive
9	Positive	27.45	Yes	Positive	26.30	Yes	Positive
10	Positive	25.07	Yes	Positive	24.80	Yes	Positive
11	Positive	28.10	Yes	Positive	27.15	Yes	Positive
12	Positive	29.83	Yes	Positive	33.00	Yes	Positive
13	Positive	37.59	Yes	Positive	26.20	Yes	Positive
14	Positive	25.48	Yes	Positive	27.10	Yes	Positive
15	Positive	27.43	Yes	Positive	32.60	Yes	Positive
16	Positive	35.39	Yes	Positive	29.40	Yes	Positive
17	Positive	29.26	Yes	Positive	29.30	Yes	Positive
18	Positive	28.82	Yes	Positive	36.35	Yes	Positive
19	Positive	37.21	Yes	Positive	31.60	Yes	Positive
20	Positive	32.56	Yes	Positive	29.55	Yes	Positive
21	Positive	31.50	Yes	Positive	33.95	Yes	Positive
22	Positive	40.08	Yes	Positive	36.65	Yes	Positive
23	Positive	37.51	Yes	Positive	26.45	Yes	Positive
24	Positive	26.01	Yes	Positive	27.25	Yes	Positive
25	Positive	29.12	Yes	Positive	26.10	Yes	Positive
26	Positive	26.88	Yes	Positive	32.30	Yes	Positive
27	Positive	31.41	Yes	Positive	29.70	Yes	Positive
28	Positive	30.85	Yes	Positive	33.40	Yes	Positive
29	Positive	33.11	Yes	Positive	33.95	Yes	Positive
30	Positive	27.99	Yes	Positive	27.15	Yes	Positive
31	Negative	ND	Yes	Negative	36.80	Yes	Positive
32	Negative	ND	Yes	Negative	40.25 *	Yes *	Negative
33	Negative	ND	Yes	Negative	40.40 *	Yes	Negative
34	Negative	ND	No	Inconclusive	ND	Yes	Negative
35	Negative	ND	Yes	Negative	ND	Yes	Negative
36	Negative	ND	Yes	Negative	ND	Yes	Negative
37	Negative	ND	Yes	Negative	ND	Yes	Negative
38	Negative	ND	Yes	Negative	ND	Yes	Negative
39	Negative	ND	Yes	Negative	ND	Yes	Negative
40	Negative	ND	Yes	Negative	ND	Yes	Negative
41	Negative	ND	Yes	Negative	ND	Yes	Negative
42	Negative	ND	Yes	Negative	ND	Yes	Negative
43	Negative	ND	Yes	Negative	ND	Yes	Negative
44	Negative	ND	Yes	Negative	ND	Yes	Negative
45	Negative	ND	Yes	Negative	ND	Yes	Negative
46	Negative	ND	Yes	Negative	40.30 *	Yes	Negative
47	Negative	ND	Yes	Negative	ND	Yes	Negative
48	Negative	ND	Yes	Negative	41.40 *	Yes	Negative
49	Negative	ND	Yes	Negative	ND	Yes	Negative
50	Negative	ND	Yes	Negative	ND	Yes	Negative

* Sample with Cq out of acceptance interval; ND = not detected.

**Table 4 diagnostics-14-00892-t004:** Absolute numbers and percentages of detections: true-positive, true-negative, false-positive, and false-negative rates, sensitivity, specificity, and Kappa coefficient. Results were calculated from data shown in Table 3.

Category	ABI7500			Q3-Plus		
	Detections	Total Possible	Rate (%)	Detections	Total Possible	Rate (%)
Positive	30	30	100	30	30	100
Negative	19	20	95	19	20	95
Inconclusive	1	50	2	0	50	0
False-negative	0	30	0	0	0	0
False-positive	0	20	0	1	20	5
Positive predictive value (PPV)	100%			99.45% (CI95% 96.38 to 99.92%)		
Negative predictive value (NPV)	99.8% (CI95% 98.9 to 99.9%)			100%		
Sensitivity	100% (CI95% of 88.43% to 100%)			100% (CI95% of 88.43% to 100%)		
Specificity	100% (CI95% of 82.35% to 100%)			95% (CI95% of 75.13 to 99.87%)		

## Data Availability

ADTC is the guarantor of this work and, as such, had full access to all the data in the study and takes responsibility for the integrity of the data and the accuracy of data analysis.

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
