# Peer review of "Validation of a New Duplex Real-Time Polymerase Chain Reaction for Chlamydia trachomatis DNA Detection in Ocular Swab Samples"

_diagnostics, 2024, doi:10.3390/diagnostics14090892_

Round 1
Reviewer 1 Report
Comments and Suggestions for Authors
The reviewed manuscript is dedicated to development and validation of a qPCR for detection of Chlamydia trachomatis that causes eye infection leading to blindness. The presented results are interesting for scientists, specializing on the field of molecular diagnostics and would help to introduce a simple and cost-effective assay for developing countries. However, several issues could be amended before acceptance.
Major issues:
1. Primer concentrations seems to be low at the range of 100-200 nM. Plausibly, higher concentrations would be beneficial in terms of sensitivity and robustness as any slight issues with oligonucleotide synthesis will severely affect PCR efficiency.
2. For positive and negative clinical samples, Cq ranges of human DNA are different, and the reason for it would be highly appreciated. Also, details of how these ranges were determined are necessary. This information is extremely important as it concerns sensitivity of the assay.
3. In section 2.3, the Cq range for C. trachomatis is 22–38, while in Table 2 many positive samples have much higher Cq. This discrepancy needs to be explained specifically.
4. Figure 2C — samples out of the qPCR linear range cannot be quantified.
5. Taking into account relatively high Cq values for C. trachomatis, 33-40, high frequency of positive samples seems to be suspicious and can stemmed from possible cross-contamination during sample acquisition. Have authors considered testing mock negative samples from the endemic area?
Minor issues:
1. Minor language editing is necessary for correcting typos, missing or excessive spaces, etc.
2. As far as I remember, human rRNA genes are highly multiplied sequences with several hundred copies per genome. In that light, its usage as an internal control can impede amplification of much less abundant C. trachomatis target leading to potentially lower sensitivity. To avoid this problem, a single copy gene can be used as an internal control.
3. The heading for Table 1 seems to be missing.
4. Authors are encouraged to add Cq values for LoD.
5. qPCR curves would be more readable if presented in a classical sigmoidal form rather than in a current logarithmic one.
6. Isothermal amplification methods, such as LAMP, are generally believed to be more suitable for POC testing than PCR. Thus, the discussion section would be further strengthened by comparison of the reported qPCR assay with previously published LAMP tests.
Comments on the Quality of English LanguageMinor language editing is necessary for correcting typos, missing or excessive spaces, etc.
Reviewer 2 Report
Comments and Suggestions for Authors
The develop and validate a new research duplex qPCR reaction for concomitant detection of Chlamydia trachomatis cryptic plasmid and the human 18S rRNA gene in ocular swab samples. The newly developed reaction was validated against qPCR-pre-characterized samples, showing high sensitivity and specificity.
The methodology used in this study exhibits robustness, and the results obtained are promising.
As a minor suggestion for improvement, it would be beneficial to include a concise table summarizing the qPCR conditions and a graphical representation, such as a flowchart outlining the study design and the proposed PCR procedure.
Comments on the Quality of English LanguageNA
Author Response
A flowchart outlining the study design and a new table summarizing the qPCR conditions are now presented as a graphical abstract and Table 1, respectively.
Round 2
Reviewer 1 Report
Comments and Suggestions for Authors
I greatly appreciate the author's efforts in manuscript editing and their detailed replies to questions during the review. However, a couple of questions still need to be addressed before publication:
1. “However, we did test higher concentrations such as 200-400 nM but there was no change to the assay´s sensitivity or specificity.” — The results of these experiments would be welcomed in the manuscript because they demonstrate the robustness of the designed test.
2. Authors are encouraged to add their reply to the question about possible sample cross-contamination in the manuscript. It is a crucial aspect of testing because careful precautions are necessary to avoid possible false-positive results when highly sensitive molecular assays are used. Thus, the general audience could once again be warned about issues related to sample handling and processing.
Author Response
1. “However, we did test higher concentrations such as 200-400 nM but there was no change to the assay´s sensitivity or specificity.” — The results of these experiments would be welcomed in the manuscript because they demonstrate the robustness of the designed test.
Reply. We agree with the reviewer. These data are shown as new Supplemental Figure S2.
2. Authors are encouraged to add their reply to the question about possible sample cross-contamination in the manuscript. It is a crucial aspect of testing because careful precautions are necessary to avoid possible false-positive results when highly sensitive molecular assays are used. Thus, the general audience could once again be warned about issues related to sample handling and processing.
Reply. The reviewer makes a great point. We have included the discussion regarding cross-contamination issues not only in the field during research activities, but also within the community between individuals, in Discussion section (highlighted in blue in the text).

Round 3
Reviewer 1 Report
Comments and Suggestions for Authors
Many thanks to authors for their detailed response to comments and clear answers to all questions mentioned in the review. All concerns were properly addressed, and no further changes of the manuscript are necessary for its publication.